EMBO
Molecular Medicine

# SARS-CoV-2 outbreak investigation in a German meat processing plant

Thomas Günther[1,†], Manja Czech-Sioli[2] (iD), Daniela Indenbirken[1], Alexis Robitaille[1], Peter Tenhaken[3], Martin Exner[4], Matthias Ottinger[5], Nicole Fischer[2,*], Adam Grundhoff[1,†,**] (iD) & Melanie M Brinkmann[6,7,***] (iD)

## Abstract

We describe a multifactorial investigation of a SARS-CoV-2 outbreak in a large meat processing complex in Germany. Infection event timing, spatial, climate and ventilation conditions in the processing plant, sharing of living quarters and transport, and viral genome sequences were analyzed. Our results suggest that a single index case transmitted SARS-CoV-2 to co-workers over distances of more than 8 m, within a confined work area in which air is constantly recirculated and cooled. Viral genome sequencing shows that all cases share a set of mutations representing a novel sub-branch in the SARS-CoV-2 C20 clade. We identified the same set of mutations in samples collected in the time period between this initial infection cluster and a subsequent outbreak within the same factory, with the largest number of confirmed SARS-CoV-2 cases in a German meat processing facility reported so far. Our results indicate climate conditions, fresh air exchange rates, and airflow as factors that can promote efficient spread of SARS-CoV-2 via long distances and provide insights into possible requirements for pandemic mitigation strategies in industrial workplace settings.

**Keywords** aerosol transmission; meat processing plant outbreak; SARS-CoV-2 super spreading event; viral genome sequencing
**Subject Categories** Chromatin, Transcription & Genomics; Microbiology, Virology & Host Pathogen Interaction

## Introduction

The first wave of SARS-CoV-2 infections peaked in Europe from March to mid of May 2020. Implementation of social and physical distancing measures resulted in declining infection numbers in most European countries. Currently, countries seek to implement alternative measures, for example, infection management focused on hotspots, contact tracing, and sentinel testing. Given this, it is important to immediately follow-up on local infection clusters to prevent re-emergence of large-scale community transmission as seen during the first wave of SARS-CoV-2 infections.

Transmission of SARS-CoV-2 is thought to mainly occur via respiratory uptake of droplets (van Doremalen *et al*, 2020) or aerosols. Aerosols are believed to be particularly important in cases where a single source transmits the virus to a large number of individuals, so-called super spreading events (Dyal *et al*, 2020; On Kwok *et al*, 2020; Schwierzeck *et al*, 2020; Xu *et al*, 2020; Yusef *et al*, 2020; Zhang *et al*, 2020a, 2020b). Whereas droplets typically travel no farther than 2 m, aerosols can stay in the air for prolonged periods of time and may deliver infectious viral particles substantially beyond 2 m distances, especially in indoor settings with low fresh air exchange rates (Liu *et al*, 2017; Asadi *et al*, 2019, 2020). Factors such as temperature, humidity, and air circulation are thought to significantly influence stability and transport of droplets and aerosols and consequently transmission efficiency (van Doremalen *et al*, 2020).

Meat processing plants have recently emerged as hotspots of SARS-CoV-2 around the world. This is thought to result not only from operational practices (e.g., close proximity of workers in the production line combined with physically demanding work that promotes heavy breathing), but also from sharing of housing and transportation that may facilitate viral transmission (Dyal *et al*, 2020). The requirement to operate at low temperature in an

1   Heinrich Pette Institute, Leibniz Institute for Experimental Virology, Hamburg, Germany
2   Institute for Medical Microbiology, Virology and Hygiene, University Medical Center Hamburg-Eppendorf, Hamburg, Germany
3   Health Office, Osnabrück, Germany
4   Institute of Hygiene and Public Health, University of Bonn, Bonn, Germany
5   Omikron Systems GmbH, Braunschweig, Germany
6   Viral Immune Modulation Research Group, Helmholtz Centre for Infection Research, Braunschweig, Germany
7   Institute of Genetics, Technische Universität Braunschweig, Braunschweig, Germany
    *Corresponding author. Tel: +49 40 7410 55171; Fax: +49 40 7410 53250; E-mail: nfischer@uke.de
    **Corresponding author. Tel: +49 40 48051 275; Fax: +49 40 48051 277; E-mail: adam.grundhoff@leibniz-hpi.de
    ***Corresponding author. Tel: +49 531 6181 3069; E-mail: m.brinkmann@tu-braunschweig.de
    †These authors contributed equally to this work

environment with low air exchange rates is another factor that may promote spread of the virus among workers. However, direct scientific evidence for the nature of transmission events in a meat processing plant or the role of shared housing and transportation has not been reported yet.

Here, we report a transmission cluster in a German meat processing plant in May 2020 and provide data suggesting that environmental conditions promoted viral transmission from a single index case to more than 60% of co-workers within a distance of 8 m. Viral sequence analyses revealed a previously unreported SARS-CoV-2 genotype that is not only shared by all individuals of the initial cluster, but also by samples collected shortly before a subsequent outbreak in mid-June, which represents the largest outbreak in a meat processing plant seen in Germany thus far. Our findings indicate that a physical distance of 2 m does not suffice to prevent transmission in environmental conditions such as those studied here; additional measures such as improved ventilation and airflow, installation of filtering devices, or use of high-quality face masks are required to reduce the infection risk in these environments.

# Results

We studied an outbreak in the largest meat processing plant in Germany, located in Rheda-Wiedenbrück, county of Gütersloh, state of North Rhine Westfalia (referred to as MPP-R in the following). MPP-R performs slaughter and fine processing as well as packaging of beef and pork. A second, independently operated processing plant specialized on sow deboning (MPP-D in the following) is located in Dissen (county of Osnabrück, state of Lower Saxony), approximately 30 km away from MPP-R. Due to occasional SARS-CoV-2 positive cases in the German meat industry, several state governments in Germany arranged SARS-CoV-2 PCR-based series testing of the entire staff of meat processing plants in May 2020, including MPP-D and MPP-R.

### Series of events preceding the outbreak in meat processing plant R (MPP-R)

As shown in Fig 1A, government-mandated series testing in MPP-D and MPP-R took place in the week of May 11. Test results were reported on Sunday May 17. Ninety-four out of 279 tested MPP-D employees were found to be SARS-CoV-2 positive, suggesting an ongoing outbreak among MPP-D workers. In MPP-R, only four out of a total of 6,289 employees were found to be positive.

None of the four cases in MPP-R was involved in meat processing and the cases were judged to likely be independent. On Tuesday May 19 (Fig 1B), two MPP-R workers from the early shift (referred to as cases B1 and B2 in the following) reported to the management of having had a brief contact with employees from MPP-D (D1 and D2 in the following) on Sunday May 17, both of whom had received positive test results later that day (Fig 1A). Cases B1 and B2 reported to have no symptoms.

### SARS-CoV-2 outbreaks in MPP-R during May and June 2020

B1 and B2 were tested in the company's test center on May 20 (Fig 1B). Because the contact with MPP-D workers was not classified as high risk, both continued to work. On May 21, the early shift did not work due to a holiday. Upon receiving positive test results on May 21, B1 and B2 and five workers with whom they had shared an apartment were quarantined. B1 and B2 were moved to a separate apartment, whereas their flatmates remained in their original quarters. On Monday May 25, all remaining workers from the early shift ($n = 140$) were tested. Test results from May 27 found 18 early shift workers to be positive. All early shift workers were immediately quarantined thereafter. Follow-up testing performed between May 27 and June 3 identified another 11 positive cases among the already quarantined workers.

Following this outbreak in May, risk- and evidence-based screening performed by health authorities, general practitioners, and the internal MPP-R test center identified increasing numbers (> 110) of positive cases across different parts of the plant in June, suggesting an ongoing and more wide-spread second outbreak event. Indeed, subsequent series testing by health authorities between June 17 and 23 identified more than 1,400 positive cases, constituting the largest outbreak in a German meat processing facility seen thus far (Fig 1C).

### Viral genotypes in the May 2020 outbreak

The timing of events suggested employees B1 and B2 as the most likely source(s) of the early MPP-R infection cluster. To further substantiate this hypothesis, we performed full viral genome sequencing of the 20 cases tested positive by May 27. In Fig 2A, we present a heat map showing positions and color-coded frequency

**Figure 1. Timeline of events.**

A  Series of events in MPP-R and MPP-D (boxes with solid and dashed outline, respectively) preceding the outbreak in MPP-R. The encounter between MPP-R and MPP-D workers which may have initiated the outbreak in MPP-R is shown in the gray box to the right. * quarantine in MPP-D involved all positive cases ($n = 94$) as well as those among the 185 workers with negative test results who had been directly employed in meat processing. Negatively tested employees with other roles (e.g., administrative or security staff) were not quarantined.

B  Events in MPP-R during the outbreak in late May. The three consecutive days during which the index case B1 worked in the early shift and thus the time period during which work-related exposure may have occurred is highlighted in blue.

C  Events in MPP-R during June. Risk- and evidence-based sampling during early June was performed by health authorities, general practitioners, as well as the internal test center from MPP-R. While we do not have exact information regarding the total number of positive cases for this time period, we have indicated a minimum incidence number above the timeline (110) based on publicly available reports from the local health authorities (https://www.kreis-guetersloh.de/aktuelles/corona/pressemitteilungen-coronavirus/). Boxes below the timeline mark positive cases from internal MPP-R testing that were subjected to viral genotyping. Cases designated as pork processing workers were directly associated with meat processing (deboning, shearing, packaging), while internal employees denote individuals working in areas such as the convenience food section, technical operation, or occupational safety.

Data information: Saturdays, Sundays, and holidays are shown as gray numbers across all panels.

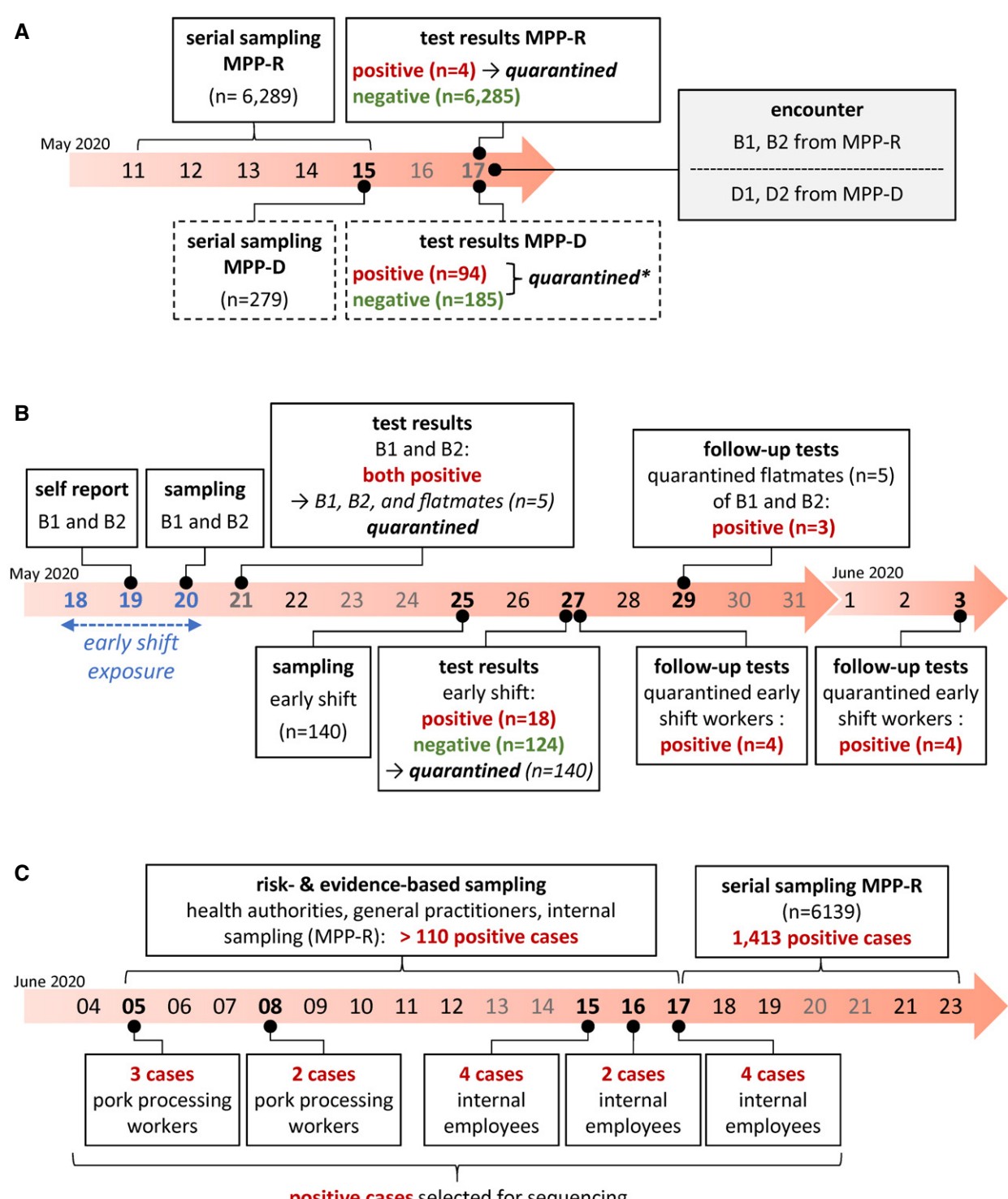

**Figure 1.**

values of nucleotide deviations from the Wuhan SARS-CoV-2 reference strain.

A total of eight exchanges were found with near 100% frequency across all samples. A search against 56,366 full-length sequences available through GISAID identified six of these mutations to be commonly present in the 20C clade of SARS-CoV-2, a branch which accounts for approximately 17% of all SARS-CoV-2 sequences deposited in GISAID at the time of this writing. Interestingly, however, we did not find GISAID entries sharing the two remaining mutations (marked with asterisks in Fig 2A; see Appendix Table S1 and Appendix Fig S3 for further details). Combined, the eight mutations therefore represent a novel sub-branch within the 20C clade

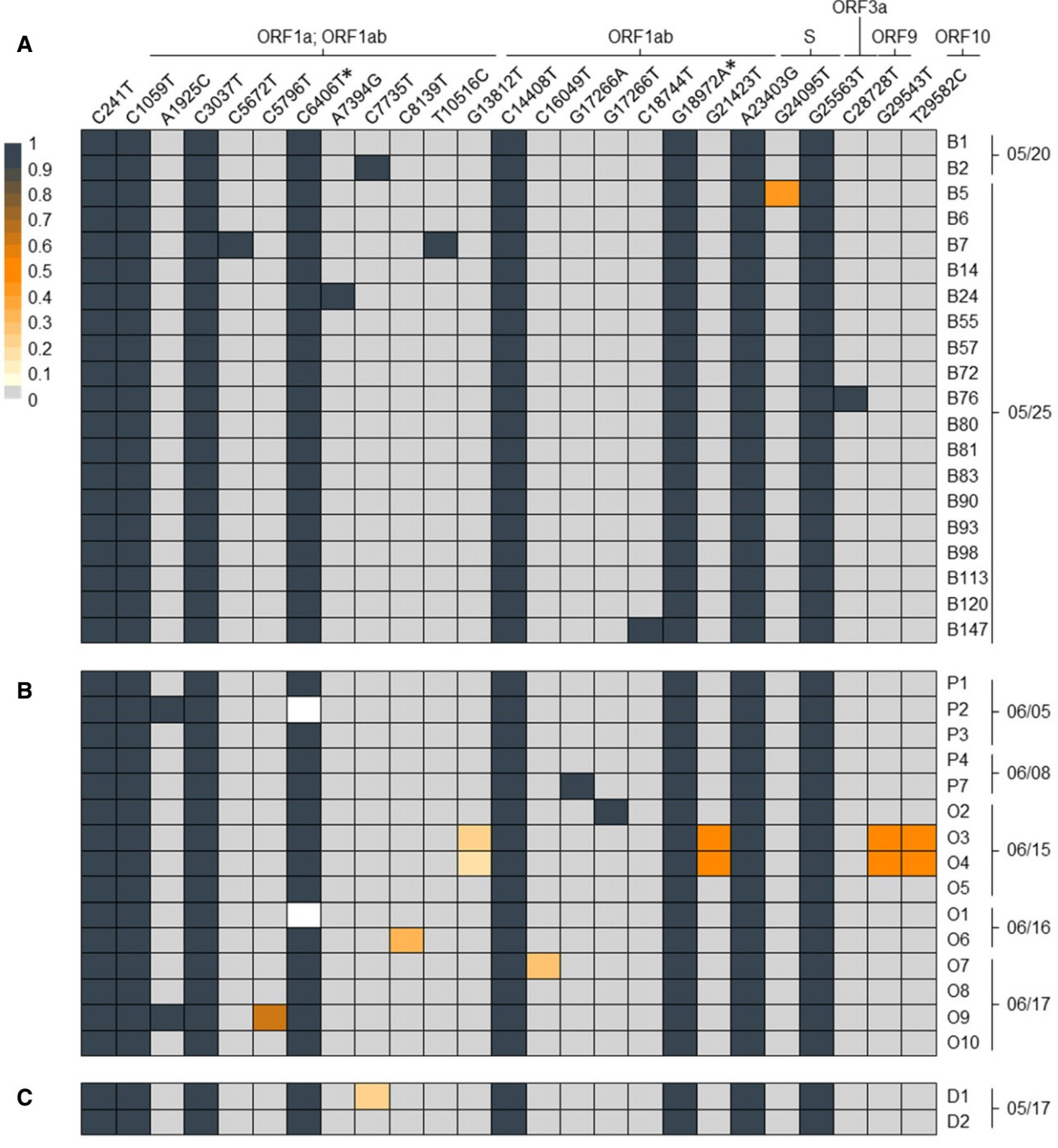

**Figure 2. SARS-CoV-2 genotypes.**

A–C The heat map shows the position (left to right), identity (top row), and frequency (color code) of variant nucleotide positions detected by SARS-CoV-2 full genome amplicon sequencing in (A) 20 samples of MPP-R workers tested positive on May 20 or 25, (B) 15 samples of MPP-R workers tested positive between June 5 and 17, and (C) two workers from MPP-D who may have transmitted the virus to cases B1 and/or B2. Individual collection dates are shown to the right of each sample. Variant sequences are given relative to the Wuhan reference strain NC_045512. The two silent mutations which define the prototype of the investigated outbreak are marked with an asterisk. Frequencies below 100% mean that only a fraction of viral genomes shows nucleotide variations, indicating the presence of viral intra-host sub-populations. White rectangles denote nucleotide positions which were not covered by amplicon-seq reads within the respective sample. For nucleotide positions in coding regions, the corresponding viral ORF(s) are shown above the variant position. Variants without such information are located in non-coding regions. Absolute values for variant frequencies and amino acid changes associated with nucleotide variants, along with identifiers of entries which were submitted to GISAID are provided in Appendix Table S1.

that defines the prototypical viral genome signature of the infection cluster (submitted to GISAID, accession number 476705, strain id NRW-MPP-1). Whereas the B1 sequence is an exact prototype representative, we find an additional nucleotide exchange at 100% frequency (C7735T) in B2. The fact that this mutation is absent from the other samples rules out B2 as a possible source of the cluster with near certainty. Another six cases also exhibit a single additional nucleotide exchange that is not shared with any other sample.

Taken together, these observations suggest prototype virus transmission by B1 as the common source of infection in the cluster. Given the overall scarcity of non-prototypical nucleotide variants, the presence of additional exchanges most likely resulted from ongoing viral mutagenesis in a subset of newly infected individuals. However, the sequencing data alone cannot rule out the formal possibility that at least some of these variants represent independent infection events.

### Potential transmission routes in the May 2020 outbreak

Given the above, we investigated potential transmission routes between the suspected index case B1 and the other employees within the cluster. The universal point of potential contact among all cases was work in the early shift of the beef processing plant. The shift comprises 147 individuals, most of whom work at fixed positions in a conveyor-belt processing line. The processing line occupies an elongated area approximately 32 meters (m) long and 8.5 m wide (see floor plan in Appendix Fig S1A). Quarters of beef enter at one end of the line (referred to as proximal in the following) and are processed as they move in longitudinal direction across the room, until cuts are finally packaged near the far end of the line (referred to as distal in the following). Eight air conditioning units placed near the ceiling in the proximal half of the room constantly cool the air. Fans project the air in a lateral direction, either directly from frontal openings in the unit or via perforated hoses mounted underneath the ceiling (see schematics in Appendix Fig S1A–C), effectively sectioning the room into zones in which air is perpetually recirculated.

While data protection regulations do not allow us to indicate the precise position of the suspected index case, we can disclose that the individual occupied a fixed station within the proximal half of the room. Figure 3A furthermore indicates the position of 86 employees relative to the suspected index case, along with test results and (where available) viral genotypes (see Appendix Table S1 for details). These 86 individuals include all employees with fixed work positions in the proximal half of the processing line ($n = 56$), 22 employees with fixed work stations in adjoining areas and estimated average location of eight employees who typically move around the room during the shift (marked with an asterisk in Fig 3A). While we do not have precise location information for the remaining 60 early shift workers (only one of whom tested positive on June 6), all of these individuals occupied fixed stations within the distal half of the processing area.

The map in Fig 3A immediately suggests a spatial relationship between the location of the suspected index case and the SARS-CoV-2 positive workers. As shown in the distance matrix in Fig 3B (see also Appendix Table S2), the probability for spatial overrepresentation of positive cases is highly significant and reaches a maximum (*P*-val 2.33E-05) within a radius of 8 m (referred to as 8 m area hereafter; note that while the 8 m maximum reflects statistical significance of overrepresentation, infection rates *per se* are higher in closer proximity to the index case).

In addition to work area locations, we were provided with information on apartments ($n = 11$), bedrooms ($n = 16$), and carpools ($n = 9$) shared among workers from the early shift. In Fig 3C, we show a statistical overrepresentation analysis of positive cases in shared units. The 8 m area around the index case is shown for comparison. Positive rates were statistically significant only for a single shared apartment and associated carpool (a1 and c3), and a shared bedroom (r5). The fact that 5 of 7 members in a1/c3, and 2 out of 3 members in r5 have fixed work stations within the 8 m area (Appendix Table S3), however, suggests that high infection rates in these units primarily reflect the number of group members who work in close proximity of the index case, rather than resulting from independent infection chains within the units themselves. This hypothesis is furthermore supported by a general positive correlation (average Pearson correlation coefficient $r = 0.67$) between unit infection rate and percentage of unit members working in the 8 m area (Appendix Fig S2). Hence, while some secondary infections may have occurred within apartments, bedrooms, or carpools, our collective data strongly suggest that the majority of transmissions occurred within the beef processing facility, with case B1 being at the root of the cluster.

### Viral genotypes in infection events before and after the May 2020 outbreak

The timeline in Fig 1 suggests a continuous transmission chain between the initial cluster in May and the larger outbreak among MPP-R employees in June. We therefore determined viral genotypes in samples from 15 MPP-R employees collected during the early phase of the second outbreak. These included five samples from pork deboning workers who had tested positive on June 5 (P1, P2) or 8 (P3, P4, P7), and ten samples from employees with various internal roles tested positive between June 15 and 17 (O1–10). As shown in Fig 2B, all samples exhibit the dominant signature mutations defining the prototypic sequence from the early infection cluster in May. Additional nucleotide variants with frequency values between ~ 20 and 100% were present in seven of the samples. Among the latter, two pairs (P2/O9, O3/O4) exhibited variant patterns which suggest that one of the employees had infected the other, or that both had acquired the virus from an individual not included in our sequencing regimen. Finally, we sought to evaluate whether the two hallmark mutations at position 6,406 and 18,972 may have emerged in the index case B1 or may have been already present in the ancestral virus. We therefore acquired samples from the two MPP-D workers (D1 and D2) who may have been in contact with B1 and B2 on May 17. As shown in Fig 2C, both MPP-D workers share the prototype sequence seen in B1. Of note, D1 additionally exhibits the same mutation (C7735T) that differentiates the genotype in case B2. Hence, since D1 sequences show this mutation with a frequency of ~ 20%, it is possible that this individual may have been the common source of infection, passing on the prototypic sequence to case B1 and a variant genome to B2.

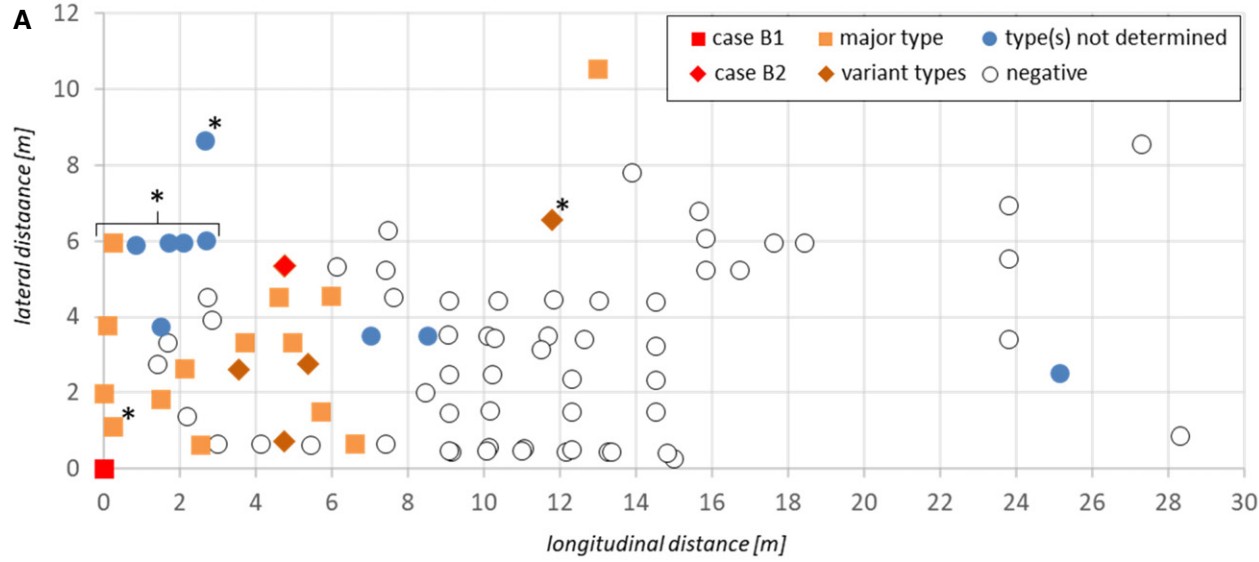

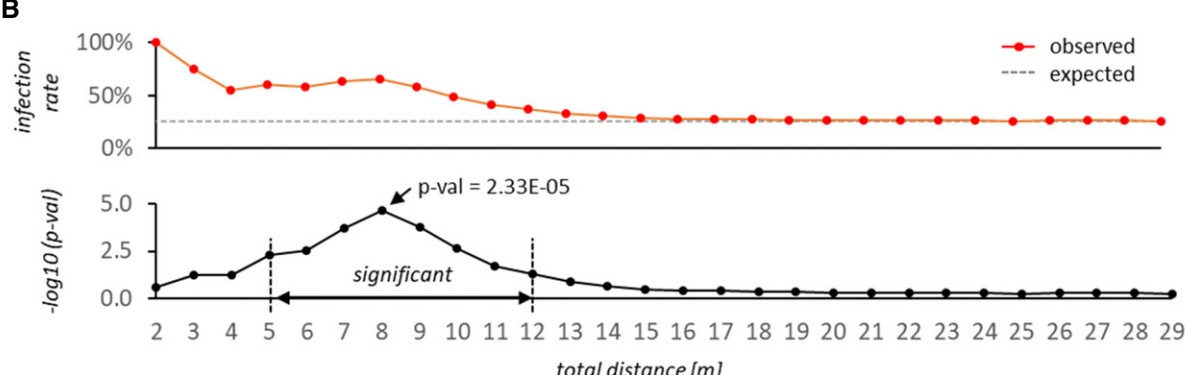

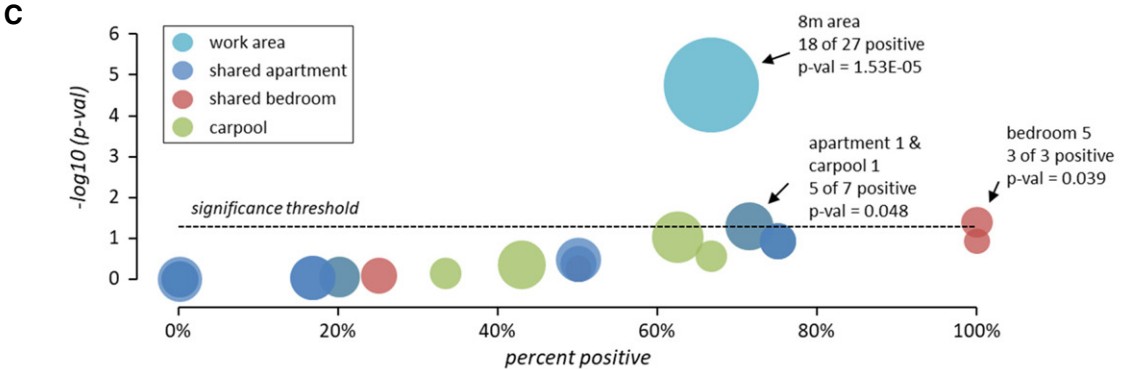

**Figure 3. Workplace location and infection events in the beef processing plant.**

A  Distance (in meters) of PCR-tested workers from the suspected index case B1 at the workplace. For workers without fixed position in the beef processing plant (marked with an asterisk), coordinates indicate estimated average location during the early shift. Squares and diamonds denote prototype or variant SARS-CoV-2 genotypes, respectively. Filled blue circles denote cases for which viral genomes were not sequenced (i.e., workers tested positive after 5/25/2020). Positive test dates and genotypes are given in Appendix Table S1.

B  *Top panel*: Observed accumulated percentage of positive cases (*red line*) within the indicated distance from the suspected index case. The gray dashed line shows the average infection rate that would be expected for a random spatial distribution of positive cases. *Bottom panel*: −log$_{10}$ P-value for the frequency of accumulated positive cases within the given distance being significantly higher than expected based on a random spatial distribution of positive cases (see Materials and Methods and Appendix Table S2 for numeric values and further information on P-value calculation). Only employees with fixed work positions were included in the calculation.

C  Values on the *x*-axis show infection rates among members of shared apartments, bedrooms, or carpools. Values on the *y*-axis reflect −log$_{10}$ P-values for the hypothesis that the infection rate within a given unit is higher than expected based on a random distribution of positive cases among all workers sharing one or more unit (see Materials and Methods and Appendix Table S3 for numeric values and further information on P-value calculation). Infection rates and P-values associated with the 8 m work area around the index case (see panels A and B) are shown for comparison. Bubble sizes indicate the total number of individuals within each unit or area. All data points with significant P-values (≤ 0.05) are labeled with unit or area id, positive and total number of associated individuals, and P-value.

## Discussion

Our results collectively point toward a super spreading event in the MPP-R beef processing plant that originated from a single employee. Our findings suggest that the facilities' environmental conditions, including low temperature, low air exchange rates, and constant air recirculation, together with relatively close distance between workers and demanding physical work, created an unfavorable mix of factors promoting efficient aerosol transmission of SARS-CoV-2 particles. It is very likely that these or similar factors are also responsible for current worldwide ongoing outbreaks in other meat or fish processing facilities. The recurrent emergence of such outbreaks suggests that employees in meat or fish processing facilities need to be frequently and systematically screened to prevent future SARS-CoV-2 outbreaks. Furthermore, immediate action needs to be taken to quarantine all workers in a radius around an infected individual that may significantly extend beyond 2 m. Importantly, while we observed transmission in a ~ 8 m area, exact transmission distances are likely to vary substantially depending on facility layout and operation conditions. Additional studies are therefore required to determine the most important parameters which may be altered to lower infection risk, for example via optimization of airflow or ventilation conditions.

In contrast to work-related exposure, shared apartments, bedrooms, or carpools appear not to have played a major role in the initial outbreak described in this study. Nevertheless, we cannot exclude the possibility that shared living quarters or work rides may have contributed to viral transmission in context of the second, larger outbreak in June 2020. Our genotyping results are fully compatible with the hypothesis that this second outbreak was seeded by cases related to the initial cluster. We point out, however, that we have no information regarding the frequency of the NRW-MPP-1 genotype within the broader population. While the genotype had not been deposited in GISAID as of May 19, and while we thus far have not seen it in our own sequencing of approximately one thousand infected individuals from the Hamburg metropolitan area (Grundhoff and Fischer, unpublished), it is formally possible that NRW-MPP-1 may already have been more broadly distributed in the general population of the Gütersloh district in early June. In this context, it should also be noted that much of the production line workforce in meat processing facilities (including the majority of workers described here) is provided by external sub-contractors, potentially creating lines of transmission that interconnect facilities. It is therefore conceivable that NRW-MPP-1 is a genotype that may already have been particularly abundant among contractor employees. Given the large number of infected individuals in the second outbreak, it is likely that, by now, the NRW-MPP-1 genotype will have spread to the local population. It will therefore be difficult to retrospectively distinguish between the above possibilities. We therefore suggest that, in addition to frequent PCR testing across facilities, a subset of positive samples should be routinely subjected to viral genotyping to allow molecular tracing.

In conclusion, this study indicates that transmission of SARS-CoV-2 can occur over distances of at least 8 m in confined spaces under conditions of relatively low air exchange rates and high rates of recirculated unfiltered air. The significance of this study is imminent for the meat and fish processing industry but might well reach beyond these industries, and points to the importance of air quality/flow in confined spaces to prevent future super spreading events.

Finally, we would like to point out important limitations of our study: Firstly, all data on workers, including work place location and sharing of apartments or transport, were provided by the employer (MPP-R). While the employer readily answered all our requests and we have no reason to doubt the accuracy or completeness of the provided information, we did not perform independent validation of this information. Secondly, while the authors performed a site visit, environmental conditions such as airflow direction or speed were only investigated qualitatively. Hence, while we believe this does not affect our major conclusions, our investigation should not be considered an epidemiological study.

## Materials and Methods

### On-site conditions

Work conditions were inspected during an on-site visit of the beef processing plant of MPP-R during operating hours on June 2, 2020. Information on housing, commuting, and workplaces of the workers was provided by MPP-R.

### Sample collection and SARS-CoV-2 RT-PCR

Oropharyngeal swab samples from workers in MPP-R were taken in the company's SARS-CoV-2 test center and analyzed by RT-qPCR in an accredited laboratory (Labor Kneißler GmbH & Co.KG, Burglengenfeld, Germany). Oropharyngeal swab samples from workers in MPP-D were taken by public health authorities in Osnabrück, Germany. 35 samples from MPP-R and two samples from MPP-D were sent to the University Medical Center Hamburg-Eppendorf for independent SARS-CoV-2 RT-qPCR confirmation and virus genome sequencing. For RT-qPCR, samples were mixed 1:1 with Roche PCR Media kit buffer (Roche, USA). SARS-CoV-2 qPCR was performed as described (Corman & Drosten, 2020; Puelles *et al*, 2020).

Clinical samples from the University Medical Center Hamburg Eppendorf were processed according to protocols approved by Ethics Committee of the City of Hamburg (PV7306; WF026/13). The study and all measures taken to comply with current data protection and ethics regulations were registered with the ethics committee of the University of Bonn, North Rhine Westphalia, Germany, and agreement for publication within the framework of disease control, outbreak management and quality assurance was requested. The committee issued a statement of no objection to publish the study under reference number 337/20.

### SARS-CoV-2 amplicon sequencing and bioinformatic analysis

Sample preparation for SARS-CoV-2 amplicon sequencing was performed as described (Quick, 2020) with modifications (Pfefferle *et al*, 2020). Primer sequences are provided in the Appendix Table S4. Samples were sequenced on an Illumina MiSeq using 500-cycle MiSeq v2 reagent kits (Illumina). All samples were sequenced twice (including independent cDNA synthesis and library preparation reactions) to exclude the possibility of variant frequencies resulting from random amplification artifacts. Except for sample B14 (in which one sequencing reaction was excluded due to insufficient quality), reported variant frequencies reflect the average

values from independent replicates. Bioinformatic analysis was performed as described (Pfefferle *et al*, 2020), (see Appendix Table S5 for details on the amplicon-Seq statistics), with the following modifications: Input thresholds were set to at least 10 variant supporting reads with a minimum base quality of 30 (-C10 -q30). Only high confidence variants present in > 20% of reads in at least one individual sample were included and annotated using ANNOVAR 16. Minor frequency variants resulting in frameshift, stopgain, or startloss were excluded.

### Comparison of viral genotypes with GISAID database entries

We performed a blast search of the prototypical NRW-MPP-1 genotype identified in this study against all 56,366 sequences deposited in GISAID as of July 6, 2020. None of the entries contained the combination of the two nucleotide variants C6406T and G18972A that are shared across all samples investigated in our study. As shown in Appendix Table S6, a very limited number (23 out of 56,366 sequences) contain one of the two mutations. Two samples from the United States (collected on the same date as B1 and B2) also carry the variant C6406T, but additionally exhibit another 7 and 8 mutations. These samples clearly belong to a different sub-branch of clade 20C defined by a previously introduced mutation at position 27964. Twenty-one samples from the UK also contain one of the two mutations but belong to the separate clade 20B. The occurrence of these variants in different clade identifies them as homoplasies and suggests that these isolates are not closely related to the NRW-MPP-1 genotype.

### Statistical analysis

*P*-values in Fig 3B and Appendix Table S2 indicate the cumulative probability of infection rates among workers with fixed stations in the indicated distance ranges being equal or higher than observed, under the null hypothesis that the probability of any given individual being positive is independent of spatial location and reflects overall positive rates among workers with fixed stations around the index case (20 out of 78 = 25.6%; see Appendix Table S2). Similarly, for each shared unit in Fig 3C and Appendix Table S3, we calculated *P*-values for infection frequencies being equal or higher than observed among all individuals who share one or more unit (22 out of 65 = 33.8%; please note that due to data protection regulations we cannot reveal which worker IDs belong to a shared unit).

Cumulative probability mass values were calculated using the BINOM.DIST.RANGE function from Microsoft Excel for Microsoft 365 MSO (v16.0.12827.20328) with the following input values: probability *P*: average infection rate among workers with fixed stations (0.256) or workers sharing one or more unit (0.338), minimum number of successes s: observed number of positive workers in distance range or shared unit, trials *t* and maximum number of successes s2: total number of workers in distance range or shared unit.

### Description of housing conditions, work area conditions, and working conditions

#### Housing conditions

Many of the workers share apartments and those usually commute together to their workplace in vans organized by the company. The company provided us with anonymized information about the housing situation of the workers regarding information on shared apartments, bedrooms, and carpools. The largest housing unit encompassed seven workers for the initial outbreak in May in the beef processing plant (see Appendix Table S3; note that due to data protection regulations we cannot reveal which workers belong to individual shared units). In addition, we collected information about the work area and the working conditions during our on-site visit on June 2, 2020. During that visit, we visited the beef processing plant during operating hours accompanied by technical staff from MPP-R.

#### Work area conditions

The plant comprises separate areas in which slaughtering and meat processing is performed. While slaughtering takes place at ambient temperatures with higher air exchange rates, beef and pork processing are performed in rooms cooled to approximately 10°C with a high proportion of recirculated cooled air. The beef processing plant has a size of 2,800 m$^2$ and is 6.1 m high (Appendix Fig S1A and B). The entire room and the production line are cleaned and disinfected daily according to food hygiene regulations in Germany. On the day of the on-site visit, the temperature in area 1 and 2 in the beef processing plant ranged between 9.5–10.7°C and between 5.4–8.7°C in area 3. Relative humidity was measured to be 34% right below the cooling fans in area 1 and 68% in the remaining part of area 1, and between 67–71% in areas 2 and 3 (Appendix Fig S1A).

Cooling fans are cooling recirculated air without filters (C1–8). C3–8 are connected to a perforated hose directed toward area 3 whereas C1 and C2 lack a hose. C1 and C2 turn on only when temperatures rise above 10°C. Cooled air is expelled through the hall up to approximately 12 m. Cooling fans 3–8 are operating permanently and are expelling cooled air through attached perforated hoses. The air exchange rate for the entire beef processing plant is < 1. This means that it takes more than 1 h to have the air replaced by fresh air. Specification of the cooling fans is as follows: Manufacturer: Guenther AG & Co. KG, Fuerstenfeldbruck, Germany, Model: S-GGHF 50 Hz, Type 050.1E/17-AS, capacity 18.6 kW, airflow 6,440 m$^3$/h, air throw 37 m, dimensions: Length 1,363 mm, Height 747 mm, Depth 713 mm.

#### Working conditions

The workers in the beef processing plant that are working on the platform (proximal side) and the connected processing line (starting in area 1 and ending in the middle of area 2) are trained for specific cuts and therefore have fixed workplaces (Appendix Fig S1). Hence, workers could be traced in detail during their working hours. While 5–6 workers handle the beef quarters entering the plant on the platform and prepare them for cutting, the quarters are then translocated onto three conveyor-belt processing lines where 24–25 workers separate the meat from the bones. Next, finer cuts are performed (shearing) by 26–27 workers. Toward the distal part of the plant as well as in area 3, the beef is packed into vacuum packaging (Appendix Fig S1B). While the production line workers have fixed workplaces, the supervisory staff has flexible workplaces and commutes within the beef processing plant.

Shifts in the beef processing plant change once per day. The staff for early and late shifts are provided by two independent sub-contractors and hence no staff is exchanged between the shifts. A shift has two 30 min breaks and one break of 1 h. During breaks, the workers from a shift visit the canteen. Workers do not have fixed seats in the canteen. Supervisory staff does not spend the break times together

with the production line workers. The supervisors do not share housing or transport facilities with the production line workers.

**Measures implemented by MPP-R during SARS-CoV-2 pandemic**

With the onset of the SARS-CoV-2 pandemic, additional preventive measures for the production staff were imposed by MPP-R. The company adhered to the recommendations of the relevant occupational Health and safety guidelines (BGN "Ergänzung der Gefährdungsbeurteilung im Sinne des SARS-CoV-Arbeitsschutzstandards Branche Fleischwirtschaft"). Additional measures were developed and implemented by the company. Hygiene regulations like hand hygiene and one-way traffic in hallways were reinforced, and an internal multi-lingual information campaign was enrolled to raise awareness for prevention and self-detection of early COVID-19 symptoms. A body temperature thermo scanner was set up to check all employees' body temperature entering the building. Workers have been made aware of the company's SARS-CoV-2 test center and were motivated to report any events where they see themselves being at risk. Specific workplace assessments were performed to decipher possibilities to extend distances between workers. Simple one-layer face masks were made compulsory. Regulations were in place to prohibit rotation between working places for the workers. Measures in the canteen were imposed to reduce physical contact and to enforce that workers would spend their break times exclusively with workers from their own shift. Since the outbreak of the pandemic, the company managed to prevent intra-company infection chains until the event described in this paper. The implementation of the measures was audited on May 15 and May 29 by unannounced inspections of the Occupational Health and Safety Experts of the competent authority and on May 20 by the Occupational Health and Safety experts of the "Berufsgenossenschaft Nahrungsmittel und Gastgewerbe". The company had set up their own test center for PCR-based SARS-CoV-2 testing in early May 2020. In the SARS-CoV-2 test, center trained staff takes oropharyngeal swab samples from workers and other staff. The samples were analyzed by RT-qPCR in an accredited laboratory (Labor Kneißler GmbH & Co.KG, Burglengenfeld, Germany). Staff was tested based on self-reported symptoms, possible contacts to other infected persons, returning to work after more than 96 h absence from work, or based on risk-assessment of possible workplace contacts.

# Data availability

Virus sequence data have been deposited at European Nucleotide Archive, ENA, https://www.ebi.ac.uk/ena/; Accession number: PRJEB40387. Virus sequences are also available at GISAID, www.gisaid.org, see Appendix Table S1 for the individual accession link information.

**Expanded View** for this article is available online.

## Acknowledgements

We thank Gereon Schulze Althoff, Dirk Moormann, Cathrin Becker, Uwe Golling, Holger Scholze, and Dennis Junkmann from Tönnies Lebensmittel GmbH & Co. KG, Rheda-Wiedenbrück, Germany, for sharing information necessary for the outbreak investigation and the possibility to visit the site of the outbreak. We thank Johannes Knobloch and Martin Aepfelbacher, Institute for Medical

**The paper explained**

**Problem**
Originating from an index case that had self-reported previous contact with potentially SARS-CoV-2 infected persons, we describe the first cluster of SARS-CoV-2 transmission in a meat processing plant in a confined working area. This infection cluster preceded a large SARS-CoV-2 outbreak in the same meat processing plant shortly thereafter involving 1,413 SARS-CoV-2 positive tested employees.

**Results**
By performing a multifactorial analysis of housing and commuting parameters along with spatial and climate conditions in the work area and viral genome sequencing, this study provides evidence that transmission occurred likely via airborne transmission of SARS-CoV-2 over long distances in a confined area of the meat processing plant. Physical work and relatively low fresh air exchange rates together with continuous recirculation of cooled air may have favored the transmission of SARS-CoV-2 among employees.

**Impact**
Our study implicates that common operational conditions in industrial meat processing plants promote the risk of SARS-CoV-2 super spreading events. Additional measures such as improved ventilation, optimized airflow management, installation of filtering devices, or the use of high-quality face masks are required to reduce the infection risk in these environments.

Microbiology, Virology and Hygiene, UKE, and Thomas Schulz, Institute of Virology, Hannover Medical School, for helpful discussions on the manuscript. We also greatly appreciate the support of and valuable discussions with Anna Kebschull, Landrätin Osnabrück, Germany. We thank the county's health authorities involved in the outbreak management for their support. This work is supported by the Helmholtz Association, Germany (W2/W3-090). MMB is supported by the SMART BIOTECS alliance between the Technische Universität Braunschweig and the Leibniz Universität Hannover, an initiative supported by the Ministry of Science and Culture (MWK) of Lower Saxony, Germany. AG is supported by the Federal Ministry of Health, Germany (HPI-COVID-19). Open access funding enabled and organized by Projekt DEAL.

## Author contributions

TG, AG, MO, NF, and MMB designed the study. TG, AG, ME, MO, NF, and MMB performed literature search. AG, MO, NF, and MMB wrote the manuscript. TG, MC-S, DI, AG, ME, MO, NF, PT, and MMB collected the data. TG, AR, and AG performed bioinformatic data analysis. TG, AG, MO, NF, and MMB generated the figures and tables. PT reviewed the manuscript.

## Conflict of interest

The authors declare that they have no conflict of interest.

## For more information

GISAID Initiative; https://www.gisaid.org

ENA Browser—European Nucleotide Archive; https://www.ebi.ac.uk/ena

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
