## [Review Process File · EMBO Molecular Medicine]

SARS-CoV-2 outbreak investigation in a German meat processing plant

Thomas Günther, Manja Czech-Sioli, Daniela Indenbirken, Alexis Robitaille, Peter Tenhaken, Martin Exner, Matthias Ottinger, Nicole Fischer, Adam Grundhoff, and Melanie Brinkmann

DOI: [10.15252/emmm.202013296](https://doi.org/10.15252/emmm.202013296)

Corresponding author(s): *Melanie Brinkmann (m.brinkmann@tu-bs.de)*, *Adam Grundhoff (adam.grundhoff@leibniz-hpi.de)*, *Nicole Fischer (nfischer@uke.de)*

Review Timeline:	Submission Date:	15th Aug 20
	Editorial Decision:	15th Sep 20
	Revision Received:	23rd Sep 20
	Accepted:	25th Sep 20

Editor: Zeljko Durdevic/Céline Carret

Transaction Report:

15th Sep 2020

Dear Prof. Brinkmann,

Thank you for the submission of your revised manuscript to EMBO Molecular Medicine and for your patience while we retrieved the last report. We have now finally received the enclosed two reports on your article and as you will see, both referees are supportive of publication. I am pleased to inform you that we will be able to accept your manuscript pending the following final amendments:

Please address the minor comments from both referees and we would encourage you to add a phylogenetic tree as suggested by referee 1.

Please provide a point-by-point letter INCLUDING my comments as well as the reviewer's reports and your detailed responses to their comments (as Word file).

***** Reviewer's comments *****

Referee #1 (Remarks for Author):

This is a really nicely executed study on an outbreak of SARS Coronavirus 2 in a German meat processing plant. The work is meticulous and really nicely executed. The paper is clear and well written and, in a world, flooded with epidemiological reports of COVID outbreaks this is one of the best. The only real minor issue is the some of the phrasing tend to lead the reader and I would prefer that the authors have a more impartial tone and allow the reader to develop the concepts within the paper. Whilst I agree this is a super shedding event, I think the title could be adapt to more representative of the study. Which is understanding an major outbreak of SARS Coronavirus 2 in a closed system, where super shedding may be the key mechanism. Additionally, although it may seem unnecessary, I think the more "genomic" of us would like to see a phylogenetic tree to better visualise this outbreak into the broader context of the viral diversity (this can be incorporated into figure 2). Other than that, excellent piece of work and provides a really unique insight into the disease epidemiology. Well done!

Referee #2 (Remarks for Author):

Review

EMM-2020-13296 "Investigation of a superspreading event preceding a large meat processing plant-related SARS-Coronavirus 2 outbreak in Germany"

The work describes a large multifactorial investigation of an outbreak in the largest meat processing complex in Germany. Analysis is done with data on timing of infection events, spatial relationships between workers, climate and ventilation, living quarters sharing and full viral genome sequences.

Very thoroughly depicted and explained time lines of events. Reasonable analysis, discussion and conclusions of the D1, D2, B1 and B2 viral genetic data. Very well described spatial correlations and risk analysis.

Comments:

Please clarify if only the positive workers from plat D were quarantined, as it appears form Figure 1A. Did the 185 negative workers continue plant operations?

What is the difference between pork processing workers and internal employees (Figure 1C) Figure 1C. The information from the official reports (from the hyperlink) of the local health authorities appears to add up to more than 200 cases over the state period, please clarify or correct the >110 positive cases count.

Please add a sentence in the amplicon sequencing and bioinformatics analysis section of Materials and Methods to define what was achieved as average NGS coverage and what was considered insufficient coverage (as pointed out to be the reason for the white boxes in Figure 2B).

Requested final amendments (EMBO MM – Celine Carret)

Please address the minor comments from both referees and we would encourage you to add a phylogenetic tree as suggested by referee 1.

We have addressed the minor comments from both referees (see below) and have added a phylogenetic tree as supplementary figure S3. We would like to use this opportunity to thank both reviewers for their time to review our manuscript and for their insightful comments.

****** Reviewer's comments ******

Referee #1 (Remarks for Author):

This is a really nicely executed study on an outbreak of SARS Coronavirus 2 in a German meat processing plant. The work is meticulous and really nicely executed. The paper is clear and well written and, in a world, flooded with epidemiological reports of COVID outbreaks this is one of the best. The only real minor issue is the some of the phrasing tend to lead the reader and I would prefer that the authors have a more impartial tone and allow the reader to develop the concepts within the paper. Whilst I agree this is a super shedding event, I think the title could be adapt to more representative of the study. Which is understanding an major outbreak of SARS Coronavirus 2 in a closed system, where super shedding may be the key mechanism. Additionally, although it may seem unnecessary, I think the more "genomic" of us would like to see a phylogenetic tree to better visualise this outbreak into the broader context of the viral diversity (this can be incorporated into figure 2). Other than that, excellent piece of work and provides a really unique insight into the disease epidemiology. Well done!

Thank you very much for your appreciation of our work and your time for reviewing it. We have adjusted the wording of some paragraphs as per your suggestion and would like to propose the following new title:

SARS-CoV-2 outbreak investigation in a German meat processing plant

As suggested, we have also included a phylogenetic tree as supplementary figure S3.

Referee #2 (Remarks for Author):

Review

EMM-2020-13296 "Investigation of a superspreading event preceding a large meat processing plant-related SARS-Coronavirus 2 outbreak in Germany"

The work describes a large multifactorial investigation of an outbreak in the largest meat processing complex in Germany. Analysis is done with data on timing of infection events, spatial relationships between workers, climate and ventilation, living quarters sharing and full viral genome sequences. Very thoroughly depicted and explained time lines of events. Reasonable analysis, discussion and conclusions of the D1, D2, B1 and B2 viral genetic data. Very well described spatial correlations and risk analysis.

Comments:

Please clarify if only the positive workers from plat D were quarantined, as it appears form Figure 1A. Did the 185 negative workers continue plant operations?

Of the 185 employees with negative test results, all who were working in meat processing halls (i.e. in areas handling meat - deboning, cutting, packaging) were quarantined, whereas employees that were not working in the meat processing halls such as administrative staff or security staff were not quarantined. We have added this information to the legend of Figure 1A.

What is the difference between pork processing workers and internal employees (Figure 1C) Figure 1C.

Pork processing workers are working at the conveyor belt, where they are deboning and cutting meat. The internal employees were working in other areas of the plant such as the convenience food section, technical operation, or occupational safety. We have clarified this in the legend of Figure 1C.

The information from the official reports (from the hyperlink) of the local health authorities appears to add up to more than 200 cases over the state period, please clarify or correct the >110 positive cases count.

The number of infected persons refers to the official report of June 17 2020 from which it can be read that at this time, 114 SARS-CoV-2 infected persons were confirmed (Kreis Gütersloh/District of Gütersloh), with only a few persons in the general population who had no relation to MPP-R. An exact number of the few persons in the general population cannot be deduced from the report dated 17th of June and previous reports.

For this reason we give an approximate number of >110.

Please add a sentence in the amplicon sequencing and bioinformatics analysis section of Materials and Methods to define what was achieved as average NGS coverage and what was considered insufficient coverage (as pointed out to be the reason for the white boxes in Figure 2B).

We thank the reviewer for this valuable suggestion and added a table, appendix Table SVI, in which we provide all Amplicon Sequencing statistics, including read count, average coverage, stdev and percentile covered.

White boxes indeed denote nucleotides positions without coverage. We changed the sentence in Figure legend 2B, now reading:

- *White rectangles denote nucleotide positions which were not covered by amplicon-seq reads within the respective sample.*

The authors performed the requested changes.

Corresponding Author Name: Prof Dr Melanie Brinkmann; Prof Dr Nicole Fischer; Prof Dr Adam

Manuscript Number: EMM-2020-13296